# Comparison of new psychiatric diagnoses among Finnish children and adolescents before and during the COVID-19 pandemic: A nationwide register-based study

David Gyllenberg[1,2,3]*, Kalpana Bastola[1,2], Wan Mohd Azam Wan Mohd Yunus[1,4], Kaisa Mishina[1,5], Emmi Liukko[2], Antti Kääriälä[2], Andre Sourander[1]

1 Department of Child Psychiatry and INVEST Research Flagship Center, University of Turku and Turku University Hospital, Turku, Finland, 2 Finnish Institute of Health and Welfare, Helsinki, Finland, 3 Department of Adolescent Psychiatry, University of Helsinki and Helsinki University Central Hospital, Helsinki, Finland, 4 Faculty of Social Sciences and Humanities, Universiti Teknologi Malaysia, Malaysia, 5 Department of Nursing Science, University of Turku, Turku, Finland

* david.gyllenberg@utu.fi

**Data Availability Statement:** Aggregated data used in the analyses are made freely available at https://github.com/davgyl/covid_psyserv (doi: 10.

## Abstract

### Background

Coronavirus Disease 2019 (COVID-19) restrictions decreased the use of specialist psychiatric services for children and adolescents in spring 2020. However, little is known about the pattern once restrictions eased. We compared new psychiatric diagnoses by specialist services during pandemic and pre-pandemic periods.

### Methods and findings

This national register study focused on all Finnish residents aged 0 to 17 years from January 2017 to September 2021 (approximately 1 million a year). The outcomes were new monthly diagnoses for psychiatric or neurodevelopmental disorders in specialist services. These were analyzed by sex, age, home location, and diagnostic groups. The numbers of new diagnoses from March 2020 were compared to predictive models based on previous years. The predicted and observed levels in March to May 2020 showed no significant differences, but the overall difference was 18.5% (95% confidence interval 12.0 to 25.9) higher than predicted in June 2020 to September 2021, with 3,821 more patients diagnosed than anticipated. During this period, the largest increases were among females (33.4%, 23.4 to 45.2), adolescents (34.4%, 25.0 to 45.3), and those living in areas with the highest COVID-19 morbidity (29.9%, 21.2 to 39.8). The largest increases by diagnostic groups were found for eating disorders (27.4%, 8.0 to 55.3), depression and anxiety (21.0%, 12.1 to 51.9), and neurodevelopmental disorders (9.6%, 3.0 to 17.0), but psychotic and bipolar disorders and conduct and oppositional disorders showed no significant differences and self-harm (−28.6, −41.5 to −8.2) and substance use disorders (−15.5, −26.4 to −0.7) decreased in this period. The main limitation is that data from specialist services do not allow to draw conclusions about those not seeking help.

5281/zenodo.6803574). Individual-level nationwide data cannot be made freely available in line with the data approval granted by the Finnish Institute for Health and Welfare (decision number THL/1745/6.02.00/2021) that can be contacted for data permits (up-to-date contact persons: https://thl.fi/en/web/thlfi-en/statistics-and-data/data-and-services/research-use-and-data-permits/contact-details).

**Funding:** The Academy of Finland Special funding for research on COVID-19 epidemic and the mitigation of its effects (335690, awarded to PI DG, https://www.aka.fi/en/). Inequalities, Interventions and a New Welfare State [INVEST] research flagship (308552, awarded to PI AS, https://invest.utu.fi/). The funders had no role in study design, data collection and analysis, decision to publish, or preparation of the manuscript

**Competing interests:** The authors have declared that no competing interests exist.

## Conclusions

Following the first pandemic phase, new psychiatric diagnoses in children and adolescents increased by nearly a fifth in Finnish specialist services. Possible explanations to our findings include changes in help-seeking, referrals and psychiatric problems, and delayed service access.

---

## Author summary

### Why was this study done?

- Healthcare emergencies and conflicts pose real threats to the availability of psychiatric services for children and adolescents.

- Previous studies compared limited pandemic and pre-pandemic periods or they did not cover the whole spectrum of psychiatric or neurodevelopmental disorders.

- This study was done to provide a comprehensive overview of the patterns of new diagnoses among children and adolescents during the pandemic.

### What did the researchers do and find?

- This study compared the predicted and observed new diagnoses from March 2020 to September 2021 based on pre-pandemic data and analyzed data from 2017 to 2021 among approximately 1 million children and adolescents per year.

- No significant changes in diagnoses were seen during the first phase of the pandemic, from March to May 2020, but new diagnoses increased by 18.5% from June 2020 to September 2021.

- Particularly high increases were observed in females, adolescents, and those living in areas with the highest Coronavirus Disease 2019 (COVID-19) rates. The highest increases in diagnoses related to eating disorders, depression and anxiety disorders, and neurodevelopmental disorders.

### What do these findings mean?

- The rapid increases in primary and secondary diagnoses by specialist services after the first 3 months of the pandemic raises concerns about gaps in services, accessibility, and continuity after the pandemic.

- Changes in supply and demand can pose serious challenges for psychiatric services, as they adapt to provide efficient services for children and adolescents.

- These data emphasize the importance of preparing for sudden changes in service use due to healthcare emergencies or crises, including the ongoing pandemic and the current conflict in Europe.

## Introduction

Several studies in Europe showed mental health symptoms increased among children and adolescents after the start of the Coronavirus Disease 2019 (COVID-19) pandemic [1–3], but service use reduced, according to a systematic review [4]. Detailed information is vital if we are to mitigate the mental health consequences of the COVID-19 pandemic in these patients. This needs to include changes in the predicted versus observed use of psychiatric services and the demographic and diagnostic groups that are most affected by the pandemic. Because of increasing new psychiatric diagnoses already before the onset of the pandemic [5], these potential trends need to be taken into account.

Pandemic stressors and restrictions, such as school closures and limited gatherings and movements, could be associated with increased psychiatric problems among children and adolescents at a later stage [6]. One meta-analysis looked at 29 published and unpublished studies covering 80,979 children and adolescents from January 2020 to March 2021. It reported that the global prevalence of clinically elevated depression and anxiety symptoms doubled in these age groups during the first year of the pandemic [7]. In contrast, a systematic review showed that the use of psychiatric services by children and adolescents decreased by an average of 28% in the early phase of the pandemic in spring 2020. That study also reported even larger decreases in presentations to emergency psychiatric hospitals [4]. Another study from the United States reported that the proportion of weekly emergency department visits among adolescent (12 to 17 years) females doubled for eating disorders during 2020 to 2022 compared to pre-pandemic period [8]. However, a number of issues remain unclear. First, most studies on psychiatric services only studied spring 2020 and only a few covered the first year of the pandemic, up to March 2021. These showed mixed results for different outcomes, for example, reduced mental health emergency department visits [9], increased [10] or reduced [11] referrals, and increased [12] or reduced [13] hospitalizations. Second, the pandemic and related restrictions affected different geographic areas and age groups differently [13,14], but little is known about how different sociodemographic groups used services as the pandemic continued. For example, adolescents living in Finland's capital Helsinki, and the surrounding areas, faced tighter restrictions than many other areas of Finland. Third, only a few papers have reported how different diagnostic groups used services. These include a Danish study that investigated the effect of the March 2020 lockdown. The authors reported that it only affected the numbers of children and adolescent patients who used psychoactive substances or had intellectual disabilities and pervasive and specific developmental disorders [15]. Meanwhile, a study carried out in Ontario, Canada, reported that referrals for depression, anxiety, and self-harm increased, and hyperactivity and aggressivity decreased, until January 2021 [16]. Another study reported that the number of visits remained similar across diagnostic groups in Southern Finland in autumn 2020 [11]. Fourth, the number of referrals, visits, and patients using services provides information on the demand for services. However, it does not describe the number of children and adolescents who are using services for the first time, for a new psychiatric disorder. The COVID-19 pandemic has potentially changed how people access services and what services they need. That is why it is important to quantify how many new pediatric patients require psychiatric services. Finally, there is limited data that quantifies the number of new diagnoses in psychiatric services during the COVID-19 pandemic when different populations at different time points have been advised to stay at home.

The aim of this study was to compare the number of children and adolescents with new primary or secondary diagnoses across periods of varying pandemic restrictions. We examined diagnoses from psychiatric specialist services for a wide range of psychiatric outcomes during the first phase of pandemic in March to May 2020 when the tightest restrictions were in place.

These were compared with June 2020 to September 2021 when restrictions gradually eased. Our primary aim was to see whether diagnoses for psychiatric and neurodevelopmental disorders decreased, as predicted, in March to May 2020, and increased in June 2020. Our secondary aim was to examine new diagnoses by sex, age group, and geographic areas, as restrictions were different for children versus adolescents and in the capital and surrounding areas versus rest of Finland. Our third aim was to study new diagnoses specific diagnostic groups, including self-harm cases that were presented to hospitals. The observed pandemic data were compared with our predictions, which were based on pre-pandemic data going back to January 2017.

## Methods

### Study design, setting, and time periods

This population-based study used aggregated time series data from national Finnish registers and comprised children and adolescents when they were aged 0 to 17 years, who were living in Finland between January 2017 and September 2021. The Finnish Institute for Health and Welfare (THL) granted us permission to use the data (decision number THL/1745/6.02.00/2021). The THL reviewed that the study plan is in line with the data permission and that data-security-related and other ethical issues are fulfilled. Therefore, a separate ethics approval was not needed. This study is reported in line with the Strengthening the Reporting of Observational Studies in Epidemiology (STROBE) guideline (S1 STROBE Checklist).

The time periods of interest were before the COVID-19 pandemic (January 2017 to February 2020), the first 3 months of the pandemic (March to May 2020) and the next 16 months of the pandemic (June 2020 to September 2021). From March to May 2020, pandemic-related morbidity rates were at their highest during the follow-up time, strict restrictions were in place in Finland and the Finnish Government announced a state of emergency across the country [17]. Schools and other educational institutions were closed down and most children moved to online learning. Exceptions were made for children up to the age of 9 years if their parents were essential workers. From March to April 2020, traffic and movements to and from the catchment area covered by the Helsinki University Hospital were also restricted in order to slow the spread of the epidemic to other parts of Finland [18]. During this period, the proportion of people treated in hospitals due to COVID-19 was considerably higher in the Helsinki University Hospital area compared to the rest of Finland (S1 Fig). Starting in early May 2020, these restrictive measures started to be gradually reduced, as infection rates declined, in attempt to move back to normal conditions [17]. School restrictions were re-implemented in many areas of Finland between December 2020 and April 2021 when infection rates increased again (S1 Fig). Children over 13 years of age returned to distance learning, but younger children were still able to go to school.

### Data sources

We used publicly available data from Statistics Finland to identify the annual population, stratified by sociodemographic characteristics, namely the number of children and adolescents aged 0 to 17 at risk; no restrictions were made to the definition of the at-risk population [19]. The Care Register for Health Care was used to identify new monthly diagnoses. The Register contains information on all patients admitted for specialist inpatient healthcare since 1969 and public outpatient visits since 1998 [5,20]. In Finland, public healthcare is divided into primary care and specialist services that compromise inpatient units, emergency rooms in conjunction with hospitals, and outpatient units of various medical specialties. In psychiatry units, the providers are a variety of occupations including physicians, psychologists, and nurses, but diagnoses are set by child and adolescent psychiatrists or resident physicians. The diagnostic validity

of schizophrenia [21], type I bipolar disorder [22], autism [23], ADHD [24], and Tourette's syndrome [25] in the register has been studied. The data used for the aggregated data included the personal identification number issued to all Finnish citizens and the primary and secondary diagnoses, using the International Classification of Diseases, Tenth Revision (ICD-10) codes [26]. We retrieved the aggregated data on 26 November 2021, but we excluded the data from October 2021 from the final analyses, as it was incomplete for certain geographic areas. The final included data was considered complete and full case analyses were conducted.

## Outcomes and stratifying variables

The main outcome was new monthly diagnoses of any primary or secondary psychiatric or neurodevelopmental disorder by specialist services between January 2017 and September 2021. These comprised ICD-10 codes F10-F99 and have been called any disorders for brevity. We included visits from all medical specialties where the diagnoses had been given. Secondary outcomes included the incidence of service use for any new diagnosis, stratified by male or female, 0 to 12 versus 13 to 17 years, and the Helsinki University Hospital catchment area versus the rest of Finland. In Finland, child psychiatry and adolescent psychiatry are separate specialties, and the age groups 0 to 12 years versus 13 to 17 years are typically used to determine which specialty the patient belongs to. They also included the incidence of new diagnoses for: substance use disorders (ICD-10 codes F10-F19), psychotic and bipolar disorders (F20-F31), depression and anxiety disorders (F32-F34, F38-F42, F93, F94), neurodevelopmental disorders (F70-F84, F90, F95), conduct and oppositional disorders (F901, F91.0-F91.3, F91.8-F92.0, F92.8, F92.9), eating disorders (F50), and intentional self-harm (external cause diagnoses X60-X84), in line with a previous report [5] and to provide an easily interpretable overview on most diagnostic categories. We did not separately study, e.g., new diagnoses of personality disorders (F60-F69), due to low incidence in this age group. The incidence of diagnoses was defined as any new outcome-specific primary or secondary diagnosis that a patient received during the study period. We tracked the data for each individual back to birth to make sure that we only recorded the first diagnosis for any specific disorder. For example, if a patient received a diagnosis of depression in 2017 and a diagnosis of psychosis in 2020, the data relating to those specific diagnoses were included in those 2 years as 2 separate incidents.

## Statistical analyses

For descriptive purposes, we have presented the number at risk, the number with any diagnosis and specific diagnostic groups, and the corresponding diagnoses per 1,000 at risk. The annual data are shown in tables and the monthly rates are shown as plots. We used separate time series for different outcomes and stratified variables by the numbers diagnosed and the numbers at risk for each month from January 2017 to September 2021. We wanted to estimate the predicted monthly numbers for each outcome after COVID-19 was declared a pandemic in March 2020. The first step was to fit the corresponding time series data from January 2017 and February 2020. We used negative binomial regression models appropriate for fitting count data and prediction of 2-sided errors. We entered the logarithmic value of the number at risk as the offset variable. We used input variables to account for potential seasonal variations and underlying linear trends with the calendar month as a categorical variable and the year as a continuous variable. The second step was to use the fitted models to retrieve the anticipated numbers, with 95% confidence intervals (CI), by predicting the monthly numbers, along with standard errors for data, after March 2020. Third, we calculated absolute and relative changes, by comparing the observed numbers for March 2020 onwards to the predicted numbers and their 95% CIs. After conducting these analyses for the main outcome "any psychiatric

diagnosis" in the total data, we conducted sub-analyses of any diagnosis stratified by sex, age group, and geographic area to examine whether findings were consistent in males or females or in groups differentially affected by restrictions. Finally, we conducted sub-analyses for diagnostic groups to examine whether the number of new diagnoses of particular types of disorders changed during the pandemic. In post hoc supplemental analyses, we stratified analyses by sex for outcomes that had no frequencies under 5 in the sex-specific time series. In addition to examining the monthly changes, we pooled the changes for the first phase from March to May 2020 and the following phase from June 2020 to September 2021. When plotting the monthly observed and predicted rates, we used rates instead of counts for increased interpretability and comparison across analyses. We defined statistically significant findings as changes between observed and predicted numbers that did not include any values of zero in the 95% CI. The modeling and data visualization were conducted using R software, version 3.6.3 (R Foundation, Vienna, Austria). Codes and data to reproduce the analyses are available online at https://github.com/davgyl/covid_psyserv (doi: 10.5281/zenodo.6803574). The research plan and any deviations to it are described in S1 Supporting information.

## Results

### Description of the data

The population at risk ranged from 1,037,378 to 1,035,517 children and adolescents aged 0 to 17 during 2017 to 2021. During the study years, the majority were aged from 0 to 12 years of age (range 70.3 to 72.3%) and the percentage who lived in the Helsinki University Hospital catchment area ranged from 39.1% to 40.0% (Table 1). The number using specialist services and psychiatric diagnoses for any disorder ranged from 16,637 to 16,592 during 2017 to 2020. These were predominantly for neurodevelopmental disorders (range 8,000 to 9,171) and depression and anxiety disorders (range 7,838 to 8,224) (Table 1).

### Diagnoses of any psychiatric or neurodevelopmental disorder

Fig 1A shows the incidence rates for diagnoses of any disorder by specialist services and psychiatric diagnoses during the whole follow-up period. Fig 1B shows the monthly differences between the observed and predicted rates for follow-up visits during the COVID-19 pandemic, together with the main restrictions that affected children and adolescents. During the pre-pandemic period of January 2017 to February 2020, the incidence of diagnoses was lowest during the Finnish school holiday months of July and December (Fig 1A).

The observed incidence gradually decreased from the predicted incidence during the first phase of the COVID-19 pandemic in March to May 2020, which had the strictest restrictions (Fig 1A). There was a relative decrease of −9.8% (95% CI −14.3 to −4.7) in May 2020 (Fig 1B). When we pooled the 3 months from March to May 2020, the observed and predicted numbers for diagnoses for any disorder were 4,131 and 4,320 (95% CI 4,088 to 4,552), respectively. These resulted in differences that were not statistically significant: The absolute difference was −189 patients (95% CI −421 to 43) and the relative difference was −4.4% (95% CI −9.3 to 1.1).

When we pooled the data for June 2020 to September 2021, the period after the first phase of the COVID-19 pandemic, new diagnoses were almost a fifth higher than predicted. The observed number for diagnoses for any disorder was higher than the predicted number (24,433 versus 20,612, 95% CI 19,407 to 21,817) and the absolute and relative differences were 3,821 patients (95% CI 2,616 to 5,026) and 18.5% (95% CI 12.0 to 25.9), respectively. During the less restricted summer and autumn months of 2020, namely June to November, the monthly relative difference between the observed and predicted rates ranged from a decrease of 6.3% and an increase of 10.6% (Fig 1B). Adolescents were subject to various restrictions

**Table 1. Population at risk of psychiatric and neurodevelopmental disorders in 2017–2021 and number of cases by stratifying variables and secondary outcomes.**

| | Year | | | | |
|---|---|---|---|---|---|
| | **2017** | **2018** | **2019** | **2020** | **January–September 2021** |
| | No. (%) | No. (%) | No. (%) | No. (%) | No. (%) |
| Population at risk[a] | | | | | |
| All | 1,066,261 | 1,058,091 | 1,049,057 | 1,041,526 | 1,035,517 |
| Males | 545,401 (51.2) | 541,154 (51.1) | 536,391 (51.1) | 532,412 (51.1) | 529,354 (51.1) |
| Females | 520,860 (48.8) | 516,937 (48.9) | 512,666 (48.9) | 509,114 (48.9) | 506,163 (48.9) |
| Age 0–12 years | 771,241 (72.3) | 761,691 (72.0) | 748,934 (71.4) | 737,131 (70.8) | 727,224 (70.2) |
| Age 13–17 years | 295,020 (27.7) | 296,400 (28.0) | 300,123 (28.6) | 304,395 (29.2) | 308,293 (29.8) |
| Helsinki | 416,847 (39.1) | 416,141 (39.3) | 415,232 (39.6) | 414,502 (39.8) | 413,815 (40.0) |
| Rest of Finland | 649,414 (60.9) | 641,950 (60.7) | 633,825 (60.4) | 627,024 (60.2) | 621,702 (60.0) |
| Any diagnosis by specialist services[b] | | | | | |
| All | 16,637 | 16,151 | 16,189 | 16,592 | 14,948 |
| Males | 8,721 (52.4) | 8,598 (53.2) | 8,685 (53.6) | 8,413 (50.7) | 7,290 (48.8) |
| Females | 7,916 (47.6) | 7,553 (46.8) | 7,504 (46.4) | 8,179 (49.3) | 7,658 (51.2) |
| Age 0–12 years | 10,124 (60.9) | 10,094 (62.5) | 10,210 (63.1) | 10,177 (61.3) | 9,034 (60.4) |
| Age 13–17 years | 6,513 (39.1) | 6,057 (37.5) | 5,979 (36.9) | 6,415 (38.7) | 5,914 (39.6) |
| Helsinki | 6,919 (41.6) | 6,599 (41.0) | 6,837 (42.2) | 6,857 (41.3) | 7,277 (48.7) |
| Rest of Finland | 9,718 (58.4) | 9,552 (59.0) | 9,352 (57.8) | 9,735 (58.7) | 7,671 (51.3) |
| Diagnoses by specific groups[c] | | | | | |
| Substance use disorders | 1,024 | 1,127 | 1,166 | 1,185 | 695 |
| Psychotic and bipolar disorders | 377 | 317 | 328 | 337 | 227 |
| Depression and anxiety disorders | 8,200 | 8,000 | 7,838 | 8,224 | 7,043 |
| Neurodevelopmental disorders | 8,000 | 8,441 | 8,770 | 9,171 | 8,477 |
| Conduct and oppositional disorders | 1,938 | 1,767 | 1,792 | 1,684 | 1,102 |
| Eating disorders | 717 | 769 | 803 | 964 | 857 |
| Self-harm | 480 | 572 | 610 | 675 | 388 |

[a]Percentage shows comparisons with all at risk.

[b]Percentage shows comparison to all with a diagnosis.

[c]Non-mutually exclusive groups, as the sme subject could receive multiple diagnoses and the sum of subjects exceeded 100%.

between December 2020 and the end of the study period in September 2021. The observed rates were consistently higher than the predicted rates during this period, ranging from 9.2% to 45.4% (Fig 1B).

## Results by sex, age, geographic area, and diagnostic groups

Fig 2 shows the monthly predicted and observed diagnoses, stratified by sex, age, and where the patients lived. Fig 3 shows the same data for the diagnostic groups. The absolute and relative changes for March to May 2020 and June 2020 to September 2021 are shown in Table 2.

We stratified the analyses of any disorder by sex (male versus female), age group (0 to 12 versus 13 to 17 years), or geographic area (Helsinki University Hospital catchment area versus the rest of Finland), as several restrictions were more aimed at adolescents than at children (Fig 1B) and the Helsinki University Hospital catchment area had higher COVID-19 hospital rates than the rest of the country (S1 Fig). This showed that the observed diagnoses were relatively lower than predicted among males (−7.2%, 95% CI −11.8 to −2.1) and those aged 13 to 17 years (−7.7%, 95% CI −13.6 to −1.0) during the early pandemic phase of March to May 2020. However, the observed diagnoses were higher than predicted across all strata in June

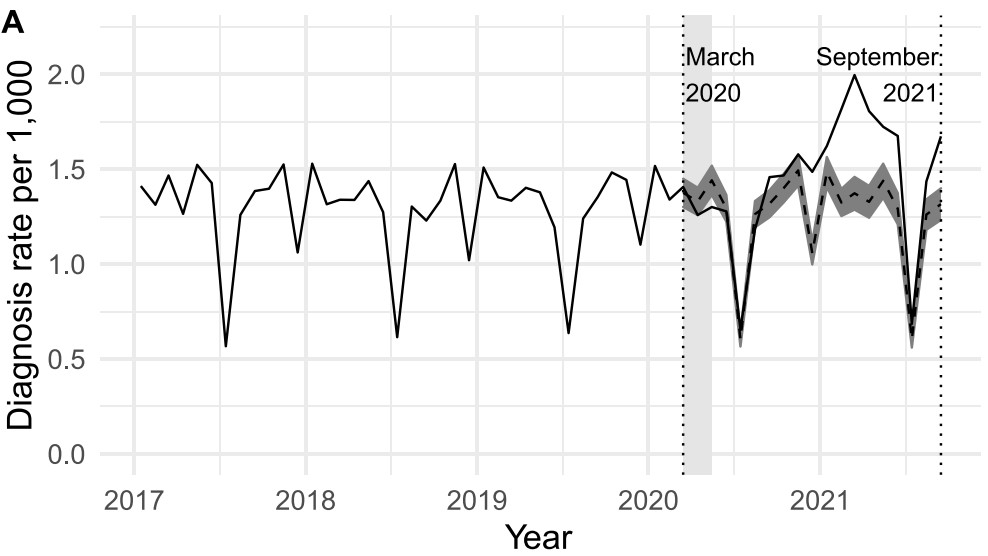

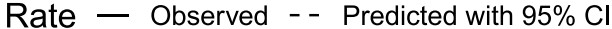

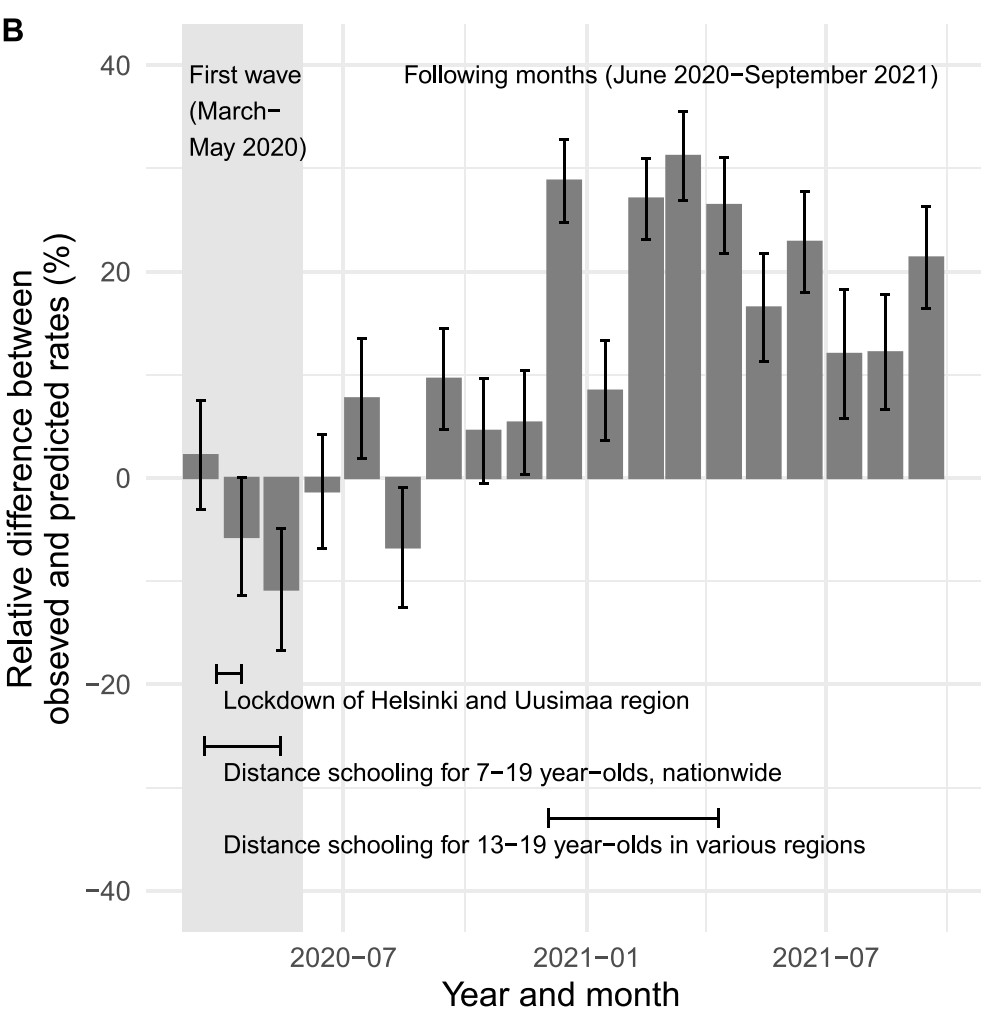

**Fig 1.** (A) Observed monthly diagnoses of any psychiatric or neurodevelopmental disorder by specialist services and psychiatric diagnoses between January 2017 and September 2021 (solid line) and predicted rates between March 2020 and September 2021 (dashed line). (B) Relative difference between observed and predicted monthly rates between March 2020 and September 2021. The light gray shaded areas represent the first phase months (March–May 2020) in both panels.

2020 to September 2021 (Fig 2 and Table 2). Table 2 shows that from June 2020 to September 2021, the largest relative differences between the observed and predicted incidences were for females (33.4%, 95% CI 23.4 to 45.2), adolescents aged 13 to 17 years (34.4%, 95% CI 25.0 to 45.3), and those living in the Helsinki University Hospital catchment area (29.9%, 95% CI 21.2 to 39.8).

In most cases, the predicted diagnoses for diagnostic groups (Fig 3) followed the similar seasonal monthly patterns as any disorders (Fig 1A). The exceptions were that substance use disorders, psychotic and bipolar disorders, and intentional self-harm did not show a prominent dip in the July school holiday months. During the first phase of the COVID-19 pandemic, in March to May 2020, the observed diagnoses were relatively lower than predicted for substance use disorders (−25.3%, 95% CI −34.5 to −13.1) and depression and anxiety disorders (−12.2%, 95% CI −18.2 to −5.4), as seen in Table 2. In June 2020 to September 2021, after the first phase of the pandemic, there was a significant increase in the observed diagnoses for eating disorders (27.4%, 95% CI 8.0 to 55.3), depression and anxiety disorders (21.0%, 95% CI 12.1 to 31.3), and neurodevelopmental disorders (9.6%, 95% CI 3.0 to 17.0) (Table 2). The increase in eating disorders and depression and anxiety peaked in September 2020, while the increase in neurodevelopmental disorders peaked later in December 2020 (Fig 3). The differences between the observed and predicted diagnoses in June 2020 to September 2021 were not statistically significant for psychotic and bipolar disorders and for conduct and oppositional defiant disorders. However, they did decrease for substance use disorders (−15.5%, 95% CI −26.4 to −0.7) and intentional self-harm (−28.6%, 95% CI −41.5 to −8.2) Table 2. For more granular insight by time, the monthly numbers are described in S2 Supporting information. In additional sex-stratified analyses for selected outcomes, patterns in males and females were similar except for depression and anxiety where females had increased observed rates, but males did not (S2 Fig).

## Discussion

This nationwide Finnish study examined new primary and secondary diagnoses for psychiatric or neurodevelopmental disorders by using population-based data from 2017 to 2021. These comprised approximately 1 million children and adolescents per year. The finding that new diagnoses were a fifth higher than our predictions, after the first phase of the COVID-19 pandemic in spring 2020, can probably be explained by changes in help-seeking, delayed treatment, and referrals and psychiatric problems.

Between June 2020 and September 2021, almost 4,000 more children and adolescents were diagnosed with psychiatric or neurodevelopmental disorders than predicted. The relative differences were as high as one-third more among females, adolescents, and those living in, or near, the capital of Helsinki, which had the highest initial COVID-19 morbidity and tightest restrictions. This high increase in new diagnoses by specialist services over such a short period has major implications for how evidence-based treatments can be delivered when services are already struggling with limited funds and human resources.

During the first phase of the pandemic, we observed an 8% reduction in diagnoses for adolescents and no difference in new diagnoses in the total data. This reduction was consistent, but lower, than the rates reported by a systematic review of psychiatric service use in 19

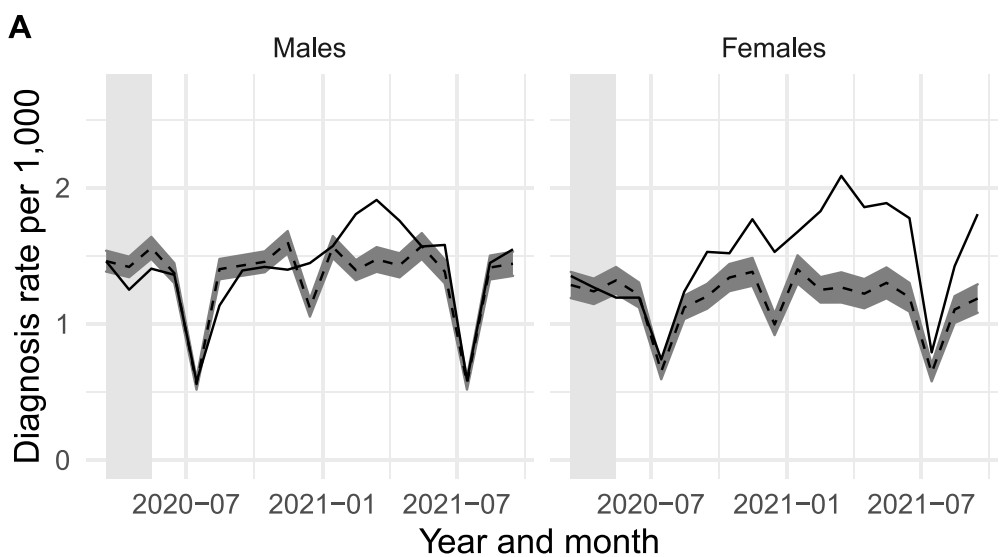

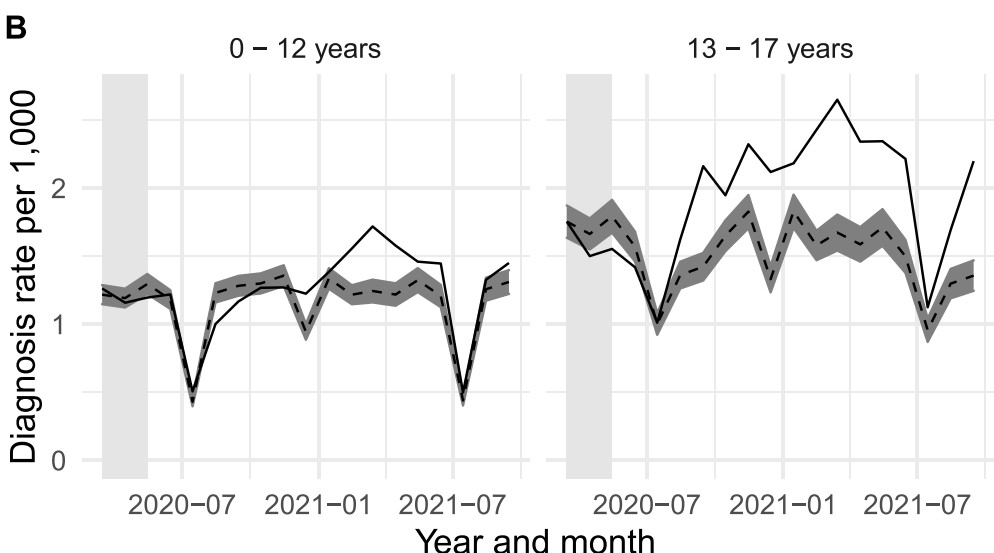

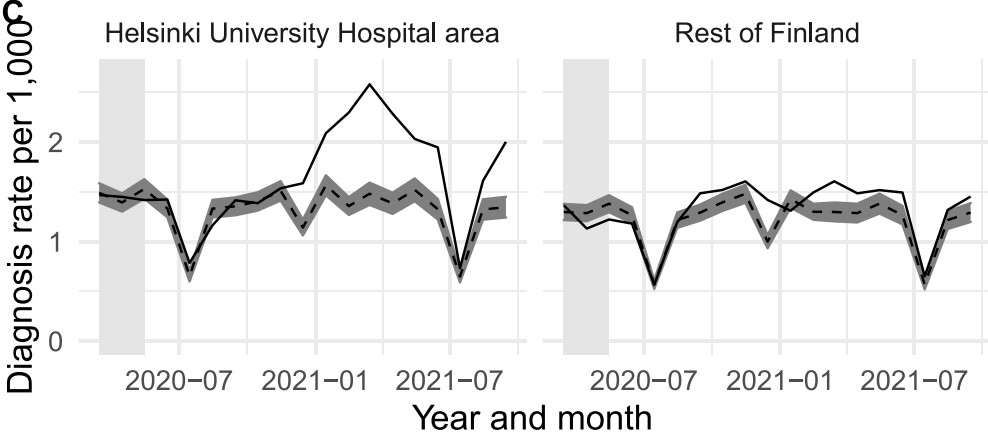

**Fig 2.** Observed (solid line) and predicted (dashed line) monthly diagnoses by specialist services and psychiatric diagnoses of any psychiatric or neurodevelopmental disorders between March 2020 and September 2021 stratified by sex (A), age group (B), and geographic area (C). The light gray shaded areas represent the first phase months (March–May 2020).

countries and regions during the same period [4]. Only a few papers have reported follow-up studies of how children and adolescents used mental health services between autumn 2020 and summer 2021 and these yielded mixed results. A study from the Republic of Ireland reported that referrals to specialist child and adolescent mental health services in September and November 2020 were 50% and 180% higher than previous years, respectively [10]. In contrast, a Finnish study reported no change in the number of patients and a decreased number of referrals to adolescent psychiatry services from June to December 2020 [11]. Our study extends previous findings by reporting new diagnoses and showing a major increase in diagnoses during 2021.

Plausible explanations of the increase in new diagnoses include delayed access to services, changes in help-seeking, and referrals and psychiatric problems. A WHO survey conducted in 130 countries between June and August 2020 showed that 72% countries reported disruption in services for children and adolescents with mental, neurological, and substance use disorders during the COVID-19 pandemic [27]. Access to care may have been disrupted due to the fear of infection of parents and caregivers causing delayed access to services [28]. This changed help-seeking and delay in treatment could have been one reason to worsening of symptoms that later in the pandemic required referral to specialist services. For example, referrals to child and adolescent mental health services recorded an initial decrease during the early phase of the pandemic followed by substantial increase as the pandemic progressed in the Republic of Ireland and the United Kingdom [10,29]. Evidence for increased symptoms has for example been provided by 2 ongoing trials in the UK: They found that the prevalence of adolescents with high depressive symptoms would have been 25% instead of 27% if the COVID-19 had not occurred [3].

Although diagnoses increased across all the demographic groups that we studied, the largest increases were among those living in the Helsinki University Hospital catchment area, adolescents, and females. During the first phase of the pandemic in spring 2020, COVID-19 morbidity and restrictions affected the Helsinki University Hospital catchment area more than other parts of Finland. This could have increased psychiatric problems among children and adolescents. Restrictions and distanced schooling had a high impact on social interactions and social isolation and loneliness increased the risk of depression and anxiety among children and adolescents [30]. However, our findings show that the major surge in observed diagnoses in the Helsinki University Hospital catchment area occurred in December 2020, which was significantly later than in other areas. This was probably because the capital had tighter initial restrictions than the rest of the country during the early stages of the pandemic. By December 2020, the difference in COVID-19 morbidity and restrictions in the Helsinki area versus rest of Finland was not as striking any more. Future studies are needed to investigate whether there was a lagged association between restrictions and increased diagnoses or whether other phenomena related to help-seeking could explain the findings.

The finding of significantly higher diagnoses among adolescents than children, 35% versus 10%, could be explained by differences in restrictions. These included distance schooling during the later phases of the pandemic, which mostly affected adolescents. It is also possible that adolescents reacted to physical distancing measures more strongly than younger children. This is because the restrictions were more likely to affect their friendship dynamics [31], especially considering that salience on peer relationship is important during adolescence [32]. Another

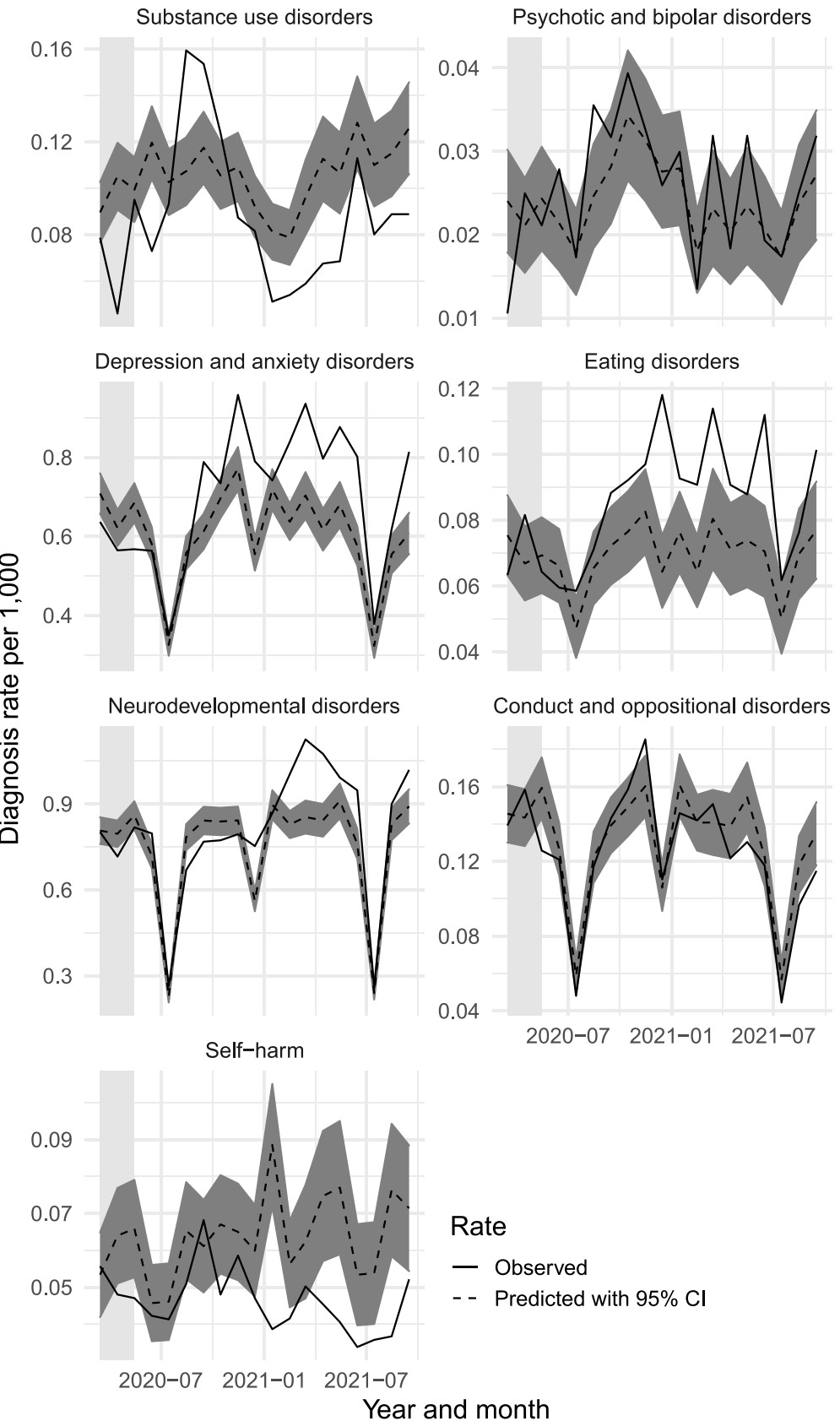

**Fig 3.** Observed (solid line) and predicted (dashed line) monthly diagnoses by specialist services and psychiatric diagnoses for the different diagnostic groups between March 2020 and September 2021. The light gray shaded areas represent the first phase months (March–May 2020). The scale of the y-axis differs for the panels to facilitate comparison between observed and predicted rates within diagnostic groups.

possible explanation for the increase in diagnoses in adolescents and females is that they tend to be more likely to be affected by depression, anxiety, and eating disorders than children and males. These findings were in line with a meta-analysis study [6] which reported that older adolescents and females had higher rates of depression and anxiety during the pandemic. A

**Table 2. Observed and predicted number of diagnoses during and after the first phase of the COVID-19 pandemic in Finland.**

| | During the first phase months (March 2020 to May 2020) | | | |
| --- | --- | --- | --- | --- |
| | Observed No. | Predicted No. (95% CI) | Absolute difference No. (95% CI) | Relative difference % (95% CI) |
| Any diagnosis | | | | |
| Males | 2,191 | 2,362 (2,239–2,485) | **−171 (−294 – −48)** | **−7.2 (−11.8 – −2.1)** |
| Females | 1,940 | 1,956 (1,810–2,102) | −16 (−162–130) | −0.8 (−7.7–7.2) |
| Age 0–12 years | 2,667 | 2,732 (2,576–2,889) | −65 (−222–91) | −2.4 (−7.7–3.5) |
| Age 13–17 years | 1,464 | 1,587 (1,479–1,694) | **−123 (−230 – −15)** | **−7.7 (−13.6 – −1.0)** |
| Helsinki University Hospital area | 1,798 | 1,830 (1,712–1,948) | −32 (−150–86) | −1.8 (−7.7–5.0) |
| Rest of Finland | 2,333 | 2,487 (2,333–2,642) | −154 (−309–0) | −6.2 (−11.7–0.0) |
| Diagnoses by specific groups | | | | |
| Substance use disorders | 229 | 306 (263–349) | **−77 (−120 – −34)** | **−25.3 (−34.5 – −13.1)** |
| Psychotic and bipolar disorders | 59 | 72 (54–91) | −13 (−32–5) | −18.5 (−35.3–10.1) |
| Depression and anxiety disorders | 1,842 | 2,099 (1,947–2,251) | **−257 (−409 – −105)** | **−12.2 (−18.2 – −5.4)** |
| Eating disorders | 218 | 221 (184–257) | −3 (−39–34) | −1.1 (−15.1–18.4) |
| Neurodevelopmental disorders | 2,433 | 2,565 (2,418–2712) | −132 (−279–15) | −5.1 (−10.3–0.6) |
| Conduct and oppositional disorders | 441 | 467 (418–516) | −26 (−75–23) | −5.5 (−14.5–5.5) |
| Self-harm | 157 | 191 (152–230) | −34 (−73–5) | −17.7 (−31.7–3.5) |
| | After the first phase months (June 2020 to September 2021) | | | |
| | Observed No. | Predicted No. (95% CI) | Absolute difference No. (95% CI) | Relative difference % (95% CI) |
| Any diagnosis | | | | |
| Males | 11,934 | 11,231 (10,584–11,877) | **703 (57–1,350)** | **6.3 (0.5–12.8)** |
| Females | 12,499 | 9,369 (8,609–10,129) | **3,130 (2,370–3,890)** | **33.4 (23.4–45.2)** |
| Age 0–12 years | 14,682 | 13,376 (12,542–14,211) | **1,306 (471–2,140)** | **9.8 (3.3–17.1)** |
| Age 13–17 years | 9,751 | 7,256 (6,713–7,798) | **2,495 (1,953–3,038)** | **34.4 (25.0–45.3)** |
| Helsinki University Hospital area | 11,133 | 8,572 (7,961–9,182) | **2,561 (1,951–3,172)** | **29.9 (21.2–39.8)** |
| Rest of Finland | 13,300 | 12,024 (11,212–12,835) | **1,276 (465–2,088)** | **10.6 (3.6–18.6)** |
| Diagnoses by specific groups | | | | |
| Substance use disorders | 1,499 | 1,774 (1,510–2,037) | **−275 (−538 – −11)** | **−15.5 (−26.4 – −0.7)** |
| Psychotic and bipolar disorders | 446 | 402 (294–510) | 44 (−65–152) | 11.0 (−12.6–51.9) |
| Depression and anxiety disorders | 11,950 | 9,878 (9,098–10,659) | **2,072 (1,291–2,852)** | **21.0 (12.1–31.3)** |
| Eating disorders | 1,466 | 1,151 (944–1,357) | **315 (109–522)** | **27.4 (8.0–55.3)** |
| Neurodevelopmental disorders | 13,495 | 12,317 (11,537–13,097) | **1,178 (398–1,958)** | **9.6 (3.0–17.0)** |
| Conduct and oppositional disorders | 2,022 | 2,107 (1,860–2,355) | −85 (−333–162) | −4.0 (−14.1–8.7) |
| Self-harm | 759 | 1,063 (827–1298) | **−304 (−539 – −68)** | **−28.6 (−41.5 – −8.2)** |

Bold text indicates 95% CIs that do not include values of zero. Predicted numbers used monthly data from January 2017 to February 2020 and were fitted with negative binomial regression models.

CI, confidence intervals; COVID-19, Coronavirus Disease 2019.

study from Ontario, Canada, also reported increased referrals to specialist children and youth services, particularly for anxiety, depression, self-harm, problem video gaming, and internet use. These were higher during March to October 2020 than the same pre-pandemic periods in 2018 and 2019 [16].

While we noted higher observed numbers of new diagnoses for several diagnostic groups than predicted during the months following the first phase of the pandemic, psychotic and bipolar disorders and conduct and oppositional disorders did not increase, and self-harm and substance use disorders decreased. The findings related to conduct and substance-related disorders are in line with previous study showing that internalizing problems, such as depression and anxiety, have increased more among adolescents during the pandemic than out-acting problems [33]. The new diagnoses of eating disorders, depression, and anxiety disorders started to increase in September 2020 and those of neurodevelopmental disorders a few months later in December 2020. The lag in diagnoses of neurodevelopmental disorders might be explained by the more extensive assessments needed for neurodevelopmental diagnoses. The number of patients or visits with self-harm and suicidal behaviors have increased during the pandemic according to a study [34] or decreased according to a multinational study [14]. This study adds to the literature by showing that new diagnoses of hospital-presenting self-harm decreased in Finland during the pandemic. The finding of no changes in new diagnoses of psychotic or bipolar disorders in Finnish specialist services during early or later phases of the pandemic is reassuring, as it shows that help-seeking and diagnostics of these severe disorders does not seem to be affected by lockdowns.

The strengths of our study include the use of national, population-based registers, which covered all specialist services from 2017 to the first 16 months of the pandemic. We also focused on patients who received new primary and secondary diagnoses during the study periods. This extensive data has enabled us to provide a comprehensive picture of the differences between predicted and observed diagnoses during the COVID-19 pandemic. The predicted number of diagnoses took into account possible previous increasing trends and seasonal variation, meaning that differences between observed and predicted diagnoses cannot be explained by, for example, previous rising service use [5]. The study had the following limitations. The registers that we used covered specialist services and we cannot draw conclusions about psychiatric symptoms among children and adolescents who did not seek help. Children and adolescents can also be diagnosed in primary care, and to a lesser extent in private care, and are accepted into specialist services upon referral if the disorder is severe. By focusing on specialist services, we were able to utilize nationwide data dating back to birth of the study subjects and thereby determine new diagnoses. However, more detailed data on diagnoses in various settings, preferably using structured clinical interviews with latest diagnostic criteria, is needed for better understanding help-seeking and referral practices. Second, we were able to provide a comprehensive overview that the pandemic associated with changed diagnoses in specialist services, but the data did not enable us to draw conclusions about what caused those changes. Future quasi-experimental observational studies, with more nuanced exposure and follow-up data, are needed to examine such patterns. Third, while the nationwide data with follow-up from birth enabled us to study the specialist service system's actualized capacity to diagnose new cases in times of changed demand, we did not have quantitative data on changes in staff capacity, waiting times, and policy changes across service systems. Multivariable time series studies with more information on these factors would be important for deepened understanding of the changes and for being prepared for new pandemics. Fourth, unfortunately, we cannot from register-based data determine whether a patient's first psychiatric diagnosis is initially incorrect and later corrected. While such a correction would not affect the main outcome "any diagnosis," it could lead to an increased number of specific diagnoses. Nonetheless,

we do not have reason to assume that such a correction process would be increased during the COVID-19 pandemic and that it could explain the increased observed rates. Fifth, the data were based on national Finnish registers and exact generalizations cannot be made to other countries. However, other studies have also shown increased referrals to mental health services after the first phase of the pandemic [10] and increased depression and anxiety, especially among females and adolescents [7,30]. This means that it is possible that these broad patterns will apply in other countries. Finally, we used census data to determine the population at risk without taking into account emigration, immigration, and prior diagnoses. However, these proportions are small in comparison to the total population: For example, the yearly immigration rate was 0.7% among 0 to 19-year-olds and has remained stable [19]. Therefore, it is unlikely that they would affect the results of changes in new diagnoses.

## Conclusions

New primary and secondary diagnoses of psychiatric or neurodevelopmental disorders by specialist services started to rise when restrictions were gradually eased and were a fifth higher than predicted by September 2021. This increase is likely explained by changes in help-seeking, referrals and psychiatric problems, and delayed access to services.

Services need to rapidly adapt to the increased number of children and adolescents with psychiatric disorders. In addition, action plans are needed to deal with future pandemics and unexpected crises, such as the Russia–Ukraine war, which will affect service use. Our results also suggest that preventive strategies are needed to tackle specific disorders such as eating, mood, and anxiety problems. They also need to target particular groups such as adolescents, females, and those living in areas more affected by the pandemic.

## Supporting information

**S1 STROBE Checklist. STROBE Statement—checklist of items that should be included in reports of observational studies.**
(PDF)

**S1 Fig. Weekly rate of COVID−19 patients treated in hospitals.**
(PDF)

**S2 Fig. Sex−stratified results for selected outcomes.**
(PDF)

**S1 Supporting information. Research plan (2021-03-14) and deviations to the plan.**
(PDF)

**S2 Supporting information. Tables showing the observed and predicted number of diagnoses by month during the COVID-19 pandemic in Finland until September 2021.**
(PDF)

## Acknowledgments

We wish to thank Dr. Pasi Moisio from the Finnish Institute for Health and Welfare for contributing to the early phase of the data retrieval process.

## Author Contributions

**Conceptualization:** David Gyllenberg, Andre Sourander.

**Data curation:** David Gyllenberg, Kalpana Bastola, Emmi Liukko.

**Formal analysis:** David Gyllenberg.

**Funding acquisition:** David Gyllenberg, Andre Sourander.

**Investigation:** David Gyllenberg.

**Visualization:** David Gyllenberg, Kalpana Bastola.

**Writing – original draft:** David Gyllenberg, Kalpana Bastola, Wan Mohd Azam Wan Mohd Yunus, Kaisa Mishina, Emmi Liukko.

**Writing – review & editing:** David Gyllenberg, Kalpana Bastola, Wan Mohd Azam Wan Mohd Yunus, Kaisa Mishina, Antti Kääriälä, Andre Sourander.

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
