## [Editor Report · Decision Letter 0]

5 Apr 2022

Dear Dr Gyllenberg, 

Thank you for submitting your manuscript entitled "Population-based comparisons of new psychiatric diagnoses among children and adolescents before and during the COVID-19 pandemic" for consideration by PLOS Medicine.

Your manuscript has now been evaluated by the PLOS Medicine editorial staff and I am writing to let you know that we would like to send your submission out for external peer review.

Please re-submit your manuscript within two working days, i.e. by Apr 07 2022 11:59PM.

Kind regards,

Beryne Odeny

PLOS Medicine

---

## [Decision Letter · Decision Letter 1]

16 May 2022

Dear Dr. Gyllenberg,

Thank you very much for submitting your manuscript "Population-based comparisons of new psychiatric diagnoses among children and adolescents before and during the COVID-19 pandemic" (PMEDICINE-D-22-01112R1) for consideration at PLOS Medicine. 

Your paper was evaluated by an associate editor and discussed among all the editors here. It was also discussed with an academic editor with relevant expertise, and sent to independent reviewers, including a statistical reviewer. The reviews are appended at the bottom of this email and any accompanying reviewer attachments can be seen via the link below:

[LINK]

In light of these reviews, I am afraid that we will not be able to accept the manuscript for publication in the journal in its current form, but we would like to consider a revised version that addresses the reviewers' and editors' comments. You will understand that we cannot make any decision about publication until we have seen the revised manuscript and your response, and we plan to seek re-review by one or more of the reviewers. 

PLOS Medicine currently has an open Call for Papers relating to a planned special issue on the COVID-19 pandemic and global mental health. Details can be found here: https://speakingofmedicine.plos.org/2022/04/05/plos-medicine-special-issue-the-covid-19-pandemic-and-global-mental-health/. Publication of the Special Issue is planned for Q4 2022 but is subject to change.

Given the content of your manuscript, the editors would like to propose that it be considered for potential inclusion in the Special Issue (subject to consideration by our panel of guest editors). Your manuscript can still be considered for publication in PLOS Medicine even if you prefer it not to be considered for the Special Issue specifically. Please indicate in your response letter whether or not you would like the manuscript to be considered for the Special Issue. 

We hope to receive your revised manuscript by Jun 06 2022 11:59PM. Please email us (plosmedicine@plos.org) if you have any questions or concerns.

We look forward to receiving your revised manuscript. 

Sincerely,

Callam Davidson,

Associate Editor 

PLOS Medicine

plosmedicine.org

Please revise your title according to PLOS Medicine's style. Your title must be nondeclarative and not a question. It should begin with main concept if possible. "Effect of" should be used only if causality can be inferred, i.e., for an RCT. Please place the study design ("A randomized controlled trial," "A retrospective study," "A modelling study," etc.) in the subtitle (ie, after a colon).

Please structure your abstract using the PLOS Medicine headings (Background, Methods and Findings, Conclusions).

Abstract Methods and Findings:

* Please quantify the main results with 95% CIs.

* Please include the important dependent variables that are adjusted for in the analyses.

Thank you for including an Author Summary. Please trim the summary such that bullet points are single sentence (maintaining 2-3 bullet points per question). Bullet points should be objective, brief, succinct, specific, accurate, and avoid technical language.

Please include continuous line numbering throughout your manuscript document.

Please place citations within square brackets and have them preceding punctuation. 

Please update your in-text citations for Supplementary Materials using the guidance found here: https://journals.plos.org/plosmedicine/s/supporting-information#loc-item-description

Please ensure that the study is reported according to the STROBE guideline, and include the completed STROBE checklist as Supporting Information. Please add the following statement, or similar, to the Methods: "This study is reported as per the Strengthening the Reporting of Observational Studies in Epidemiology (STROBE) guideline (S1 Checklist)."

Did your study have a prospective protocol or analysis plan? Please state this (either way) early in the Methods section.

Typo on page 12: Figure A should read Figure 1A.

Typo in legend for Figure 1: September 202 should read September 2021.

Please check your y-axis labels are correct – I think they should all read ‘per 1,000’ as opposed to ‘for 1,000’.

I would suggest using a solid and a dashed line to differentiate between observed/predicted rates in your figures, as the black vs grey distinction is not always obvious (though inclusion of 95% CI does make this easier). 

In the Results section, please quantify results with 95% CI. 

Please ensure claims made in the Discussion are suitably referenced (e.g. ‘restrictions were more likely to affect the kind of behaviours that teenagers engaged in, like hanging out with friends’).

The Author’s contributions, conflicts of interest statements, role of the funding source, and ethics committee approval, and data sharing statement sections can all be removed from the end of the main text (pages 21-22) – all of the information is captured elsewhere (either in the Submission form or the Methods). 

Comments from the reviewers:

Reviewer #1: Please check grammar and spellings throughout. A native English speaker may be able to help. 

Title: Appropriate

Abstract: 

Reflects the contents within the manuscript. 

Introduction:

Very well-written. Evidence before the study well-summarised and aims of the study easy to follow

Methods:

What was the rationale behind stratification by age?

I note the authors include different diagnosis as separate incidents. Can the authors preclude a wrong first diagnosis initially? I.e. is there any other reason why we might be seeing a spike, such as historically wrongly diagnosed cased being correctly diagnosed due to advancements in knowledge/reclassification of diseases.

Overall, very detailed and replicable

Results:

Given we are nearly half-way into 2022, is there still no data on the population at risk in 2021? I would recommend actual numbers replace estimated numbers where possible. 

I'm struggling to correlate the figures with the data presented in the tables. There seems to be a reduction in the number of diagnosis across Finland in 2021 compared to previous years (Table 1). I appreciate that this data is only until September, and this may be higher than values in September from previous years. Also, it may be that the under-diagnosis and over-diagnosis shown in the figures evened out long-term, but this doesn't reflect the results presented in Table 2. I'd like the data to be presented in way that is easier to correlate the table and figure findings. This can be done through legends if need be.

Discussion:

Very well written

I thoroughly enjoyed reading the paper.

Reviewer #2: The study by Gyllenberg et al. leverages nationwide Finnish data to address an extremely important topic on potential change in the incidence of psychiatric disorders among children (as diagnosed through the health care system) during the first 16 months of the COVID-19 pandemic. With a clear pattern in the results, this study adds significantly to the existing literature. I hope the following points are helpful to the authors to improve the quality of their manuscript: 

1. The decrease in incident substance use disorders and self-harm from June 2020 to September 2021 among children is interesting and for a balanced review of the results these findings should also be discussed and commented on, e.g. in the abstract and sections of discussion/conclusion in the paper. 

2. The first sentence of the introduction: „Psychiatric symptoms increased among children and adolescents after the start of the COVID-19 pandemic,1-3 but service use reduced, according to a systematic review" - needs to be clarified. Was this a global thing or occurred in a particular setting? Also, psychiatric symptoms have in many societies been increasing in children, particularly teens, during the last 10 years. I think clarifications are needed here.

3. Introduction, second to last paragraph. The rationale for the study aims lacks mentioning of how limited health consumption data are in quantifying the „need" for psychiatric services during a pandemic, when populations are advised to stay at home. Thus, estimation of incidence or new onset psychiatric disorders are limited by this factor.

4. Introduction, specific aims/last paragraph: „The aim of this study was to compare the pandemic restrictions in Finland with the number of children and adolescents who received a new primary or secondary diagnoses from psychiatric specialized services." The aim is not clearly stated, please consider to rephrase. The aim here must be to compare the incidence of primary and secondary psychiatric diagnoses in children across periods and places of varying pandemic restrictions? 

5. I noticed a few typos and grammatic errors (e.g. page 8, line 2), as well as use of the English language where I am unsure that the correct meaning is conveyed (as in point 3 above) - please review carefully.

6. Methods, first paragraph: „The Finnish Institute for Health and Welfare granted us permission to use the data (decision number THL/1745/6.02.00/2021)." What about ethics approval? Was the study exempt from such a review?

7. Methods, first paragraph: It is stated that the study used aggregated data while later (page 8, 2nd paragraph) it is stated: "The data we gathered included the personal identification number issued to all Finnish citizens and the primary and secondary diagnoses, using the International Classification of Diseases, Tenth Revision (ICD-10) codes." Please clarify.

8. Methods (page 9, paragraph 1): „We tracked the data for each individual back to birth to make sure that we only recorded the first diagnosis for any specific disorder." Was this conducted using the same data used during the study period (2017-2021)? Any information on the quality of this data source or validation studies on psychiatric disorders?

9. Methods/statistical analysis (page 9-10): The description is quite general and would benefit for more details on the approach/modeling, subgroup analyses and rationale for these decisions.

10. In a surge of a new somatic disease during a pandemic, you may hypothesize a similar surge in secondary diagnoses of anxiety and depression in this patient population. In Tables 1 and 2, it would therefore be interesting to show numbers of primary and secondary diagnoses, and preferably conduct an additional analysis contrasting expected and observed numbers by primary and secondary diagnoses.

Reviewer #3: This is a well-conducted population study on new psychiatric diagnoses among children and adolescents before and during the COVID-19 pandemic in Finland. The study design, datasets, statistical methods and analyses, and presentation (tables and figures) and interpretation of the results are mostly adequate and of a good standard. However, there are still a couple of issues needing attention.

1) The use of negative binomial regression models is appropriate and the results are mostly descriptive in terms of diagnosis rates, trends/patterns and changes before and during Covid19 pandemic. We can see the patterns of rates are very consistent in the 3 years before pandemic (2017, 2018 and 2019) so we would expect them to be the same if Covid19 never happened. The study results showed that there is a drop (not statistically significant) in the first 3 months of the pandemic and then the rates picked up later (18.4%). Then the authors concluded "This increase can probably be explained by changes in helpseeking, referrals and psychiatric problems, rather than delays in accessing services" (Page 20, conclusion). However, I am not convinced with this statement. They could all be factors for the diagnosis rate changes during the pandemic. So far the paper is mostly descriptive rather than multivariable analysis on factors contributing to the rate changes. One would imagine the delays in hospital service during the pandemic would be the most important factors of this rate change or at least a very important factor. Could authors provide the hospital service data before and during the pandemic such as consultancy appointments, closures, reduced staff capacity (%?), policy changes in hospital services, and etc? We need some solid data to support the claims of factors leading to these rate changes so that we can understand better and be prepared if it happens again.

2) The authors haven't mentioned anything about missing data. What's the quality of the data? Any missing data issues?

Reviewer #4: 

The current study examined the prevalence of new primary and secondary diagnoses from psychiatric specialized services among Finnish children and adolescents in the wake of the COVID-19 pandemic. The differences in prevalence rates were compared between when restrictions rates were the tightest (March-May 2020) and once the restrictions were eased (June 2020-September 2021) to pre-pandemic prevalence rates (January 2017-February 2020). They found a decrease in mental health services used during the first three months of the pandemic, which was followed by an increase in the 16-months that followed. The study has clear clinical implications regarding the mental health status of youth during the pandemic and the need for further resources to be devoted to prevention and intervention efforts for this vulnerable group.

There is substantial research indicating that the rates of depression, anxiety and self-harm have been on the rise among adolescents prior to the pandemic. As such, there may be an upward trend in these diagnosis during the 2017-2020 period. Was the increase in depression and anxiety after the first wave of the COVID-19 pandemic greater than this trend would suggest? The answer is likely yes; however, looking into this could provide more insight into how much the pandemic and its restrictions have negatively impacted the mental health of adolescents. 

Whether the prevalence rate increases or decreases varies considerably by disorder. If space allows, it would be interesting to provide more information on why these rates differ. For instance, any thoughts on why prevalence rates on self-harm decreased after the pandemic? Or why the prevalence of eating disorders increased?

In the discussion, the authors note that the increase in new diagnosis after the first phase of COVID-19 is likely due to changes in help-seeking, referrals and psychiatric problems rather than delayed access to services due to the pandemic restrictions. On what do the authors base this claim?

Reviewer #5: * The research question is an important one to the community of researchers in this general area.

Yes

* The results provide a substantial advance over existing knowledge, with clear implications for patient care, public policy, or clinical research agendas.

No

* Published together with an Author Summary written for general readers, the article is of interest to clinicians and policymakers who are not specialists in this topic.

Yes

* What are the main claims of the paper and how significant are they for the discipline?

The study confirms an increase in specific psychiatric diagnostic rates in adolescents starting a few months after the COVID pandemic. The study includes only first time diagnoses to determine how the pandemic is associated with mental health disorder incidence. It includes a longer time period than other studies (up until September 2021 instead of March 2021). Since these two approaches appear to be the main contributions, the findings are not particularly novel, at least compared to results from other countries. But it is beneficial to have country-specific results available, especially from those using nationally representative data with records going back to birth to determine a newly diagnosed disorder. And it is helpful to continue highlighting the importance of the associations between shocks like the COVID pandemic with mental health care utilization in youth. 

*Are the claims properly placed in the context of the previous literature? Have the authors treated the literature fairly?

yes

*Do the data and analyses fully support the claims? If not, what other evidence is required?

yes

*PLOS Medicine encourages authors to publish detailed methods as supporting information online. Do any particular methods used in the manuscript warrant such publication? If a protocol is already provided, for example for a randomized controlled trial, are there any important deviations from it? If so, have the authors explained adequately why the deviations occurred?

N/A

*Is this paper outstanding in its discipline? If yes, what makes it outstanding? If not, why not?

No. The use of a national registry is beneficial when drawing conclusions. Besides specifically focusing on newly diagnosed disorders, the study is not particularly novel in either its methodology or findings.

*Does the study conform to any relevant guidelines such as CONSORT, MIAME, QUORUM, STROBE, and the Fort Lauderdale agreement?

Yes

*Are details of the methodology sufficient to allow the experiments to be reproduced?

yes

*Is any software created by the authors freely available?

N/A

*Is the manuscript well organized and written clearly enough to be accessible to non-specialists?

yes

Abstract: 

"Changes in help-seeking, referrals and psychiatric problems are likely explanations for our findings." Versus? In the discussion it becomes clear that the other reason considered is the delay of services. This might be added to make it a less obvious/empty statement at first sight. Also, something could be added along the lines as "Considering the magnitude of excess diagnoses, changes in help-seeking, referrals and psychiatric problems are likely explanations for our findings, versus delayed access to services" 

What does specialized services mean in Finland? Do the authors mean to say specialist services? Please be explicit in what kind of services this includes, e.g. outpatient, emergency room, hospitalizations, what kind of mental health care providers (e.g. psychiatrists, psychologists, social workers, etc). Possibly in the abstract but at the very least in the main body of the text. 

Introduction: 

Comment on sentence: "However, it does not describe the number of children and adolescents who are using services for the first time, for a new psychiatric disorder". Does this sentence only apply to the study that is referred to, or all other prior studies? In the section "Why was this study done?", it is mentioned that none of the prior studies study new diagnoses. As this is one of the few potentially unique perspectives of the current study (besides the longer follow up period), this might be good to repeat in the introduction, not only related to that one study. 

Missing "is" in: "That is why it important to quantify how many new paediatric patients require psychiatric services"

The sentence "The aim of this study was to compare the pandemic restrictions in Finland with the number of children and adolescents who received a new primary or secondary diagnoses from psychiatric specialized services" does not make a lot of sense - comparing restrictions with number of people. Please rephrase. 

The introduction is thorough in its description of the prior studies, but I'm missing one CDC report that shows similar results (especially for eating disorders) in the US that the authors may wish to include (https://www.cdc.gov/mmwr/volumes/71/wr/pdfs/mm7108e2-H.pdf). 

Methods:

Is the age range 0-17 years inclusive, up to kids' 18th birthday?

The sentence: "and most children moved to online teaching" implies that the children were teaching the class. An alternative would be to use the words "online learning" or something similar.

In the US we would use the word "essential workers" instead of "vital key workers", but this might be different in Finland/Europe. 

"In early May 2020, these restrictive measures started to be gradual reduced" -> I would suggest "Starting in early May 2020, these restrictive measures were gradually reduced"

What does "at risk" mean? Is that the population without any prior diagnoses of mental health disorders? Or without a prior diagnosis for a specific disorder? Please define at first mention. 

Did the authors identify the cases in the Care Register for Health Care themselves? If so, the data needed to reproduce the results are (presumably and rightfully) not available to readers. But that means that the authors are not fulfilling the obligation to make all data fully available. 

When taking together the specific diagnostic categories, does that equal the "any disorders" category, or are there diagnoses that do not fall in any of the categories? 

Are people included if they weren't born in Finland? If so, how is determined whether they were diagnosed for the first time? Conversely, if people were born in Finland but left at some point (also temporarily), are they removed from the denominator? In other words, how do the authors deal with immigration and emigration? 

Is it possible to be diagnosed/treated for mental health disorders outside of the system, in a way that it wouldn't show up in the register, e.g. in private practices?

Am I reading it correctly that if someone receives a diagnosis of depression first and bipolar disorder later, this is counted as two separate incidents, but if this is depression first and anxiety later this would not count as two incidents? What is the justification for the combinations of the diagnostic categories? In the same vein, could an increase in incidence also mean that there are more differential diagnoses? 

It is common to use a second event with a similar diagnosis to confirm that the disorder is present, and not just one event. At least for outpatient events or emergency room visits. This could prevent incorrect outcomes based on suspected cases, disorders that need to be ruled out, etc. How do the authors currently prevent this kind of error? 

The phrase "the number with outcomes" is not clear - please specify which outcomes. 

Why did the authors not opt for methodology such as the Farrington algorithm (or adaptations) to detect whether the trends are different than the years before? Not necessarily a critique, mainly curious how these two approaches compare. I'd like to see some justification in the manuscript why the chosen approach is the most appropriate one though. 

The sentence "Statistical power was defined as the 95% CI of the predicted numbers." is an odd interpretation of statistical power and/or CIs. Please explain and rephrase. 

Results

In the sentence "We stratified the analyses of any disorder by sex (male versus female), age group (0-12 versus 13-17 years) and geographic area (Helsinki University Hospital catchment area versus the rest of Finland)", "and" should be replaced with "or", as the cases were not stratified by all conditions at the same time. 

Related to that, it would be interesting to see results from diagnostic categories by gender, sample size permitting. The CDC report mentioned earlier, with results from the US, shows increases in ED visits for eating disorders and tics only for girls and not boys. I wonder if that's the case here too, for different diagnostic categories. 

Discussion

"During the first phase of the pandemic, we observed an 8% reduction in diagnoses for adolescents and a non-significant overall differences in new diagnoses". The second part of this sentence after "and" is not clear, with respect to describing a non-significant difference (use the phrase "no difference" instead, and use singular word not plural) and what outcome is described - does "overall" here mean 0-18 years since it is juxtaposed to adolescents? Please specify. 

Add "in" to the sentence "Another possible explanation for the increase diagnoses in adolescents and females is that…". 

The sentence "These findings were in line with several reviews, which reported that older adolescents and females had depression and anxiety during the pandemic." doesn't make sense. Higher rates? 

Acknowledgements

Conceptulised -> conceptualised

[LINK]

---

## [Decision Letter · Decision Letter 2]

30 Jun 2022

Dear Dr. Gyllenberg,

Thank you very much for re-submitting your manuscript "Comparison of new psychiatric diagnoses among Finnish children and adolescents before and during the COVID-19 pandemic: A nationwide register-based study" (PMEDICINE-D-22-01112R2) for review by PLOS Medicine.

I have discussed the paper with my colleagues and the academic editor and it was also seen again by four reviewers. I am pleased to say that provided the remaining editorial and production issues are dealt with we are planning to accept the paper for publication in the journal.

[LINK]

We look forward to receiving the revised manuscript by Jul 07 2022 11:59PM.   

Sincerely,

Callam Davidson, 

Associate Editor 

PLOS Medicine

plosmedicine.org

Requests from Editors:

Data Availability Statement: "Individual-level nationwide data cannot be made freely available in line with the data approval." Please provide more information regarding the data approval (e.g., that it was granted by the Finnish Institute for Health and Welfare/Decision number).

Abstract: Please combine the Methods and Findings sections into one section, “Methods and findings”. The sentence detailing study limitations should come at the end of the 'Methods and Findings' section.

Author Summary: Please include include the headline numbers from the study, such as the sample size and key findings.

STROBE checklist: Please use section and paragraph numbers, rather than page numbers (as these are likely to be different in the published version).

S2 Figure: Please update the y-axis label to read 'per 1,000' rather than 'for 1,000'.

Figures 2 and 3: Please ensure that the y axis is identical for all panels to facilitate comparison.

Please include a Table of data by month (similar to Table 2 but providing more granular insight by time). This can be included in the Supporting Information if preferred. 

Discussion: Please expand on your interpretation of neurodevelopmental compared to psychiatric diagnoses; you note an increase in neurodevelopmental disorder diagnoses (beginning in December 2020) but this is not sufficiently discussed. 

Comments from Reviewers:

Reviewer #2: The authors have addressed most of my points, and provided satisfactory explanations if not possible to address.

I have no further comments. 

Reviewer #3: Thanks authors for their effort to improve the manuscript. I am satisfied with the response and revision. No further issues needing attention.

Reviewer #4: Thank you for the opportunity to review this interesting and important study. The authors should be commended on their willingness and ability to incorporate the feedback from reviewers. Here below are a few minor issues for the authors to consider.

How reliable are diagnoses in the registry databases? Who assigns the diagnoses in outpatient settings (child and adolescents psychiatrists, clinical psychologists, general physicians?) and what methods are typically used to derive at a diagnosis (e.g., clinical interview vs. structured clinical interview)? The diagnoses in the current study are also based on ICD-10, not the more recent ICD-11. These might be points to reflect on in the limitations section.

In the first sentence of the abstract, it is noted that there was a decrease in specialist psychiatric services for youth during the ´first phase of the pandemic´. It might be unclear to the reader what constitutes the ´first phase´. The authors should consider rewording here to clarify.

I believe there is a typo in line 274 where ´phrase´ is meant to be ´phase´.

Reviewer #5: The authors have addressed my main concerns and questions. However, during the revisions, more sentences with missing words, spelling/grammar errors, and uncommon sentence structures were introduced, and the authors need to receive input from one or more native English speakers.

[LINK]

---

## [Editor Report · Decision Letter 3]

11 Jul 2022

Dear Dr Gyllenberg, 

On behalf of my colleagues and the Academic Editor, Dr Vikram Patel, I am pleased to inform you that we have agreed to publish your manuscript "Comparison of new psychiatric diagnoses among Finnish children and adolescents before and during the COVID-19 pandemic: A nationwide register-based study" (PMEDICINE-D-22-01112R3) in PLOS Medicine.

SPECIAL ISSUE EXTENSION

Your manuscript is currently intended for publication as part of the Special Issue on the COVID-19 pandemic and global mental health. The deadline for the Special Issue is being extended to December 15 2022, with anticipated publication in Q1 2023 (subject to change dependent on submission volume). We intend to publish all papers accepted for the Special Issue simultaneously. Given that this extension was announced after you submitted your manuscript for consideration, we appreciate that you may no longer wish for your manuscript to be part of the Special Issue. If this is the case, or if you have any other questions, please feel free to contact me at cdavidson@plos.org and this can be discussed. 

PRESS

Sincerely, 

Callam Davidson 

Associate Editor 

PLOS Medicine

cdavidson@plos.org